# Use of Theory-Driven Report Back to Promote Lung Cancer Risk Reduction

**DOI:** 10.3390/ijerph182010648

**Published:** 2021-10-11

**Authors:** Luz Huntington-Moskos, Mary Kay Rayens, Amanda T. Wiggins, Karen M. Butler, Ellen J. Hahn

**Affiliations:** 1School of Nursing, University of Louisville, Louisville, KY 40292, USA; 2College of Nursing, University of Kentucky, Lexington, KY 40504, USA; mkrayens@uky.edu (M.K.R.); atwiggins@uky.edu (A.T.W.); karen.butler@uky.edu (K.M.B.); ejhahn00@email.uky.edu (E.J.H.)

**Keywords:** environmental exposure, information sharing, risk reduction

## Abstract

Report back is active sharing of research findings with participants to prompt behavior change. Research on theory-driven report back for environmental risk reduction is limited. The study aim is to evaluate the impact of a stage-tailored report back process with participants who had high home radon and/or air nicotine levels. An observational one-group pre-post design was used, with data collection at 3, 9, and 15 months post intervention. Participants from the parent study (*N* = 515) were randomized to the treatment or control group and this sample included all 87 treatment participants who: (1) had elevated radon and/or air nicotine at baseline; and (2) received stage-tailored report back of their values. Short-term test kits measured radon; passive airborne nicotine samplers assessed secondhand smoke (SHS) exposure. Stage of action was categorized as: (1) ‘*Unaware*’, (2) ‘*Unengaged*’, (3) ‘*Deciding*’, (4) ‘*Action*’, and (5) ‘*Maintenance*’. Interventions were provided for free, such as in-person radon and SHS test kits and a brief telephonic problem-solving consultation. Stage of action for radon mitigation and smoke-free policy increased from baseline to 3 months and remained stable between 3 and 9 months. Stage of action for radon was higher at 15 months than baseline. Among those with high baseline radon, observed radon decreased by 15 months (*p* < 0.001). Tailored report back of contaminant values reduced radon exposure and changed the health behavior necessary to remediate radon and SHS exposure.

## 1. Introduction

Lung cancer is the leading cause of cancer death in the United States and, globally, lung cancer deaths continue to rise [1]. A diagnosis of lung cancer is preventable primarily through the avoidance and elimination of tobacco smoke (primary smoking behavior and secondhand smoke exposure) and radon exposure (radon infiltration from soil, water supply, and decay from building materials) [2]. Public awareness regarding the combined risk of exposure to tobacco smoke and radon continues to be low. Additional avenues to raise awareness and educate the public on synergistic risk (the risk resulting from combined exposure to secondhand smoke and radon) are needed [3].

The report back process is the active sharing of findings with research participants [4]. The report back process in environmental health science is used to convey basic environmental health concepts to participants, or in this case actual testing values, and to support risk reduction and positive health behavior change. To be implemented effectively, report back requires planned and purposeful communication between researcher and participant, with a special focus on providing clear context and the clarification of concepts appropriate for varying levels of science literacy among participants [5,6,7]. In addition, the report back process must be tailored to the participant’s learning style and unique cultural factors [7]. Emerging best practices to guide the report back process stress privacy, culture, health literacy, and the use of standardized templates when providing written report back to participants [8,9]; however, very little guidance is available regarding the process of delivering report back using theory and varying formats, such as in-person, phone, and/or web-based resources.

Health behavior models can be used to guide the report back process. The Precaution Adoption Process Model is a theoretical model used to guide the study of health behaviors, including those that may be less well known, such as radon risk reduction. The Precaution Adoption Process Model outlines seven stages along a continuum toward health behavior change: (1) unaware, (2) unengaged, (3) deciding about acting, (4) decided not to act, (5) decided to act, (6) acting, and (7) maintenance [10]. This health behavior model is unique in that it includes the stages prior to the decision-making process; a time period when participants may be unaware of or unengaged by a particular environmental exposure or health hazard. Thus, the Precaution Adoption Process Model is a health behavior model that is well suited for guiding the report back of environmental exposures, such as actual radon and air nicotine values found in the indoor air environment. Previous intervention studies have used the Precaution Adoption Process Model to address radon risk reduction and increase radon precaution adoption through informational brochures and tailored video presentations [10,11].

The study of a Precaution Adoption Process Model-guided report back intervention reported here was part of a larger randomized controlled trial (RCT) with promising results on stage of action over time [12]. The larger RCT intervention tested the impact of providing free home testing for radon and air nicotine, coupled with personalized report back of actual testing values on stage of action to mitigate for radon and restrict secondhand smoke exposure in the home [12,13,14]. The report back portion of the process included a 20 to 25 min telephonic, problem-solving session using a standardized protocol to assess stage of action and deliver queries and messages tailored to the stage to guide the strategies for lowering radon (mitigation) and secondhand smoke (SHS) exposure (adopting a smoke-free home) [12]. The total sample of the parent study was 515 homeowners (treatment, *n* = 257; control, *n* = 258), with participants divided by smoking in the home using stratified quota sampling. Short- and long-term outcomes from the RCT are published elsewhere [12,13].

To address lung cancer risk, the study objective was to evaluate the impact of a stage-tailored report back process with participants who had high home radon and/or air nicotine levels. The research aims for this one-group, pre-post observational study were to: (1) evaluate the effect of a theory-driven report back intervention on the stage of action to mitigate for radon or establish a smoke-free home over the 15-month follow-up period among the treatment group subsample who received report back of at least one high baseline radon and/or air nicotine value (*n* = 87); and (2) examine the changes in observed radon and air nicotine values between baseline and 15 months post-intervention. Report back of actual values has the potential to effectively increase awareness of environmental exposures, such as radon and secondhand smoke, which are causes of lung cancer. The report back process can support positive health behavior change and reduced environmental exposure [9].

## 2. Materials and Methods

### 2.1. Design and Sample

This one-group, pre–post observational study was based on the 87 treatment group participants who had elevated radon and/or air nicotine levels at baseline and received stage-tailored report back of their values after having been randomized to the treatment group of a larger parent study. The parent study included a sample of 515 homeowners living in Central Kentucky who were randomly assigned to treatment and control groups. Kentucky has high radon risk potential [15], high smoking rates [16], and weak smoke-free protections [17]. The parent study sample was recruited from primary care clinics, a pharmacy, or at community events in person by trained staff who screened for eligibility, enrolled participants, and randomly assigned participants to a study group. Based on stratified quota sampling, half of the group reported at least one smoker in the home. Data were collected at baseline and then at 3, 9, and 15 months after the theory-driven report back intervention. The study was approved by the University of Kentucky Medical Institutional Review Board. 

### 2.2. Theory-Driven Report Back Intervention

The parent study intervention consisted of a two-part process: (1) the provision of free test kits for radon and SHS coupled with instructions for deploying the kits (verbal, written, and YouTube video links); and (2) a brief problem-solving phone consultation to report back personalized testing results. Radon measurement was completed using short-term, charcoal-based test kits (Air Chek, Inc., Mills River, NC, USA, http://www.radon.com/, accessed on 30 June 2021) with participants returning the completed kit after a 5-day testing period for laboratory analysis in a postage-paid envelope. The U.S. Environmental Protection Agency (EPA, Washington, DC, USA) recommends mitigation if radon levels are at or above 4.0 pCi/L [2,18]. Short-term, charcoal-based radon testing for less than seven days is an accurate, sensitive, practical, and low-cost method for home screening [19]. In the total sample, 6% of the radon test kits returned and analyzed were invalid due to overexposure or excessive decay [13]. SHS exposure assessment was completed using sensitive and specific passive airborne nicotine samplers [20] for the same 5-day testing period, which were subsequently mailed to the Johns Hopkins School of Public Health Environmental Health and Engineering laboratory for analysis. In the total sample, less than 1% of the air nicotine tests were invalid [13]. Control participants could request free test kits from the research team at a later date. Participants with high radon received a voucher for USD 600 to be applied towards the cost of radon mitigation.

Approximately 11 weeks after completed testing, trained interventionists conducted 20 to 25 min telephonic problem-solving sessions. The report back intervention was tailored based on baseline radon and/or air nicotine values, and the stage of action [12]. Participants with at least one high test value received results via a tailored, telephone-based, brief problem-solving report back. The brief problem solving was tailored based on the level of exposure, which was classified as one of four conditions: (1) *high radon*/*high SHS* (*radon* = ≥ *4 pCi*/*L* [2] *air nicotine* = ≥ *0.1 μg/m^3^* [20]), (2) *high radon/low SHS*, (3) *low radon*/*high SHS*, and (4) *low radon*/*low SHS*.

A scoring algorithm tool was developed to guide the delivery of the intervention. The tool contains branching logic to assist with tailoring for each stage of action and, thus, guide appropriate messaging for each unique conversation. First, the trained staff documents on the tool the answer to the following question, *Are the radon and SHS levels high?* Responses include: No, radon level is <4 pCi/L; Yes, radon level is ≥4, but <8 pCi/L, and Yes, radon level is >8 pCi/L. We based our report back of testing values on the EPA guidance [2]. As the phone conversation begins, each step is scripted, starting with “I am calling today to share the results for your radon and secondhand smoke tests and discuss your thoughts about fixing your home. Is this a good time to talk?” The scripted algorithm tool cues the staff to reinforce how the participant met them at enrollment and to schedule an alternative time and date for the next call if the current call is not feasible.

After this introduction, the participant is assessed for Precaution Adoption Process Model stage of action and introductory exposure information is shared. The participant is assessed for stage of action for radon and secondhand smoke separately on the tool. Throughout the scoring algorithm tool, the study staff are cued to encourage participants to explore their preferred next steps to take action according to their assessed stage of action. Stage-tailored queries are used to enhance self-efficacy, motivation, and behavior change. Follow-up queries incorporated the 5Rs: Relevance, Risks, Rewards, Roadblocks, and Repetition [21]. The telephone conversation is concluded with the completion of a seven-item checklist including: (1) *Provide a summary of the discussion regarding radon mitigation and secondhand smoke reduction plan*; (2) *Offer to clarify or correct any misunderstandings from summary*; (3) *Is there other info that you need?* (4) *Reminder that we will need to mail materials to help*; (5) *Reminder we’ll have a few more follow-up surveys and then will ask them to test again at the end*; (6) *Verify their address and all contact information*; and (7) *Thank them for participating and end the call* (See Appendix A). Finally, all study participants were mailed free test kits for radon and air nicotine at 15 months post report back. 

### 2.3. Measures

Demographic and personal factors collected for this study were self-reported by participants and included age, gender, race/ethnicity, education, income, and family history of lung cancer. The yes/no item *‘Do you or any other members of your household smoke cigarettes, cigars or pipes?’* assessed smoking in the home.

#### 2.3.1. Stage of Action

Stage of action related to the testing and remediation of the home for radon and SHS were categorized in the following manner: (1) ‘*Unaware*’, (2) ‘*Unengaged*’, (3) ‘*Deciding*’, (4) ‘*Action*’, and (5) ‘*Maintenance*’. To assist in the interpretation of stage of action, the original stages 3–5 (3—deciding about acting; 4—decided not to act; 5—decided to act) were combined to represent *Deciding.* The combining of stages exists in previous research studies with regard to specific health behavior(s) [22]. Using multiple items for radon and SHS with separate questions for testing and remediation, the stage of action was evaluated at baseline, 3, 9, and 15 months. Using the scoring algorithm, stage of action specific to radon was determined with the item, *Which of the following best describes your thoughts about radon mitigation for your home?* Possible responses included: *I’ve never thought about mitigating* (Stage 2); *I’m undecided about mitigating* (Stage 3); *I’ve decided I don’t want to mitigate* (Stage 4); and *I’ve decided I do want to mitigate* (Stage 5). With regard to SHS, stage of action was determined using the item, *which of the following best describes your thoughts about secondhand smoke in your home?* Responses for this item were: *I’ve never thought about asking people to smoke outside* (Stage 2); *I’m undecided about asking people to smoke outside* (Stage 3); *I’ve decided I don’t want to ask people to smoke outside* (Stage 5); and *I’ve implemented a no smoking policy in my home* (Stage 6/7). Due to the established inclusion criteria of no testing for radon in the past 2 years and the fact that air nicotine tests are not commercially available, participants could not be in *Maintenance* for radon or air nicotine testing. 

#### 2.3.2. Risk Status for Radon and Air Nicotine

Every participant in this subsample was provided a free short-term radon and air nicotine test kit in person and asked to test their homes at baseline. For the parent study, the test kits were determined to be valid only if the testing dates were recorded in a manner consistent with the prescribed study timeframe and the test kit was returned intact with all pertinent written information. The participants were categorized by their baseline test results, including ‘Tested high’, ‘Tested low’, or ‘Did not test/invalid result’. Values ≥ 4.0 pCi/L were considered ‘Tested high’ for radon using the EPA action level [2]; air nicotine values >0.1 μg/m^3^ were considered ‘Tested high’ for SHS [20]. 

### 2.4. Analytic Strategy

Study variables were summarized using means and standard deviations or frequency distributions. Due to the right-skewed distribution of radon values, this variable was log-transformed prior to analysis and the scores from baseline and 15 months were summarized using geometric means. The longitudinal analyses for the stage of action measures and testing values were conducted using repeated-measures mixed modeling, with separate models for each of the four outcomes, including stage of action for both radon mitigation and establishing a smoke-free home, and testing value for both radon and air nicotine. For each model, time was the repeated measure (i.e., the fixed effect indicating the timepoints at which the outcome was measured); the covariates included in each model were measured at baseline only. For the outcomes of stage of action to mitigate for radon and to establish a smoke-free home, the included timepoints in the models were baseline and then 3, 9, and 15 months after the tailored report back (since these were the timepoints when these outcomes were measured). For the outcomes of radon and air nicotine values, with testing completed only at the beginning and end of study, the included timepoints were baseline and 15 months after the report back. For each of the four repeated measures models, the outcome measured at 15 months was the reference value, and the corresponding outcome at the earlier timepoint(s) was compared to that (and each other, for the stage of action models) to evaluate changes over time. For all models, the included baseline covariates were age, gender, race/ethnicity, education, and family history of lung cancer. For the stage of action models, risk status, as determined by the indictor for whether the participant’s home had a high test value for the corresponding indoor air contaminant at baseline, was also included as a covariate. The model for the smoke-free home policy included only those with one or more smokers living in the home. For the two stage of action models with all four timepoints included, post hoc analysis of the repeated measures time effect in these models was accomplished using Fisher’s least significant difference procedure for pairwise comparisons. For the two models based on test values at baseline and 15 months, the comparison of the two timepoints was accomplished using the model F-test for the time effect. All analysis was completed using SAS (SAS Institute Inc., Cary, NC, USA), v. 9.4; an alpha level of 0.05 was used for inferential testing. 

## 3. Results

The average age of participants was 54 years (*SD* = 14); see Table 1. Most participants were female and white/non-Hispanic. The majority were college graduates with annual incomes of at least USD 60,000. About half of the participants lived with one or more smokers in the household, consistent with the stratified quota sampling design in the parent study. Slightly more than one-quarter had a family history of lung cancer. Consistent with the inclusion criteria for this analysis, all participants included in this study had high baseline test values for radon and/or air nicotine. More than half of participants had high baseline radon values (64%) and slightly less than half had high baseline air nicotine (47%); among the 87 participants, 8 had high baseline values for both air contaminants (9%). 

In the repeated-measures mixed model with stage of action for radon mitigation as the outcome, controlling for age, gender, race/ethnicity, education, family history of lung cancer, and baseline radon risk status, the main effect of time was significant (*F* = 43.7, *p* < 0.001; see Table 2). Relative to the maximum possible score of 5, the mean values for stage of action for radon mitigation were 1.86, 3.43, 3.56, and 2.63 at baseline, and 3, 9, and 15 months, respectively; see Figure 1. The post hoc analysis for the time effect revealed that months 3, 9, and 15 each had a higher mean stage of action score compared with baseline (*p* < 0.001 for all three comparisons), and months 3 and 9 exceeded month 15 on this outcome (*p* < 0.001 for both comparisons). The difference in stage of action for radon mitigation between months 3 and 9 was not significant (*p* = 0.53). Baseline radon risk status was significant, but none of the demographic variables included as covariates were significant.

For the stage of action for the smoke-free home policy mixed model (based on only those with smoker(s) in the home), which included the covariates of age, gender, race/ethnicity, education, family history of lung cancer, and baseline air nicotine risk status, the main effect of time was significant (*F* = 12.8, *p* < 0.001; see Figure 2). Relative to the maximum possible score of 5, the mean stage of action values for the four timepoints were 2.67, 3.59, 3.94, and 3.28, respectively. The post hoc analysis demonstrated that the means for stage of action for the smoke-free home policy at months 3 and 9 were greater than at baseline (*p* < 0.001 for both comparisons), and both month 3 and month 9 exceeded month 15 in this outcome (*p* ≤ 0.04 for both comparisons). The differences between baseline and month 15 and between months 3 and 9 were not significant (*p* ≥ 0.07 for both). Baseline SHS risk status was significant in the model, but none of the demographic covariates were significant.

The repeated-measures model based on the observed log radon values at baseline and 15 months, which included only those with an elevated baseline radon value at or above 4 pCi/L, demonstrated a significant time effect (*p* < 0.001; see Figure 2); none of the demographic covariates in the model were significant (see Table 2). The baseline geometric mean was 7.69, compared with a geometric mean of 3.52 at month 15. Of the 54 participants with high radon who received report back specific to mitigation, 7 (13%) reported at the conclusion of the study (15 months) that they had mitigated. The baseline and 15-month raw radon values (i.e., untransformed) for these 7 homes are shown in Figure 3. There was a reduction in radon levels for all 7 homes post-mitigation, although one 15-month value remained above the EPA action level of 4 pCi/L (i.e., Participant 1, with the highest baseline radon level).

The repeated measures model based on the observed air nicotine values at baseline and 15 months, which included only those with a baseline air nicotine level > 0.1 μg/m^3^, did not demonstrate a significant time effect (*p* = 0.27; see Figure 2). The mean air nicotine values for the two timepoints were 3.82 and 2.14 for baseline and month 15, respectively. Though the model did not demonstrate a significant change in air nicotine level over time, the indicator for college graduates was significant (see Table 2); this finding indicates that those with a post-secondary degree had lower air nicotine levels in their homes, averaged over the two timepoints (*p* = 0.03). Other covariates in the model were not significant. Finally, of the 41 participants who had high air nicotine at baseline and received report back of their air nicotine values, only 12 had high air nicotine values at 15 months (i.e., >0.1 μg/m^3^). The remaining 29 participants in this group (70% of all of those with high air nicotine at baseline) had an observed air nicotine below the cutoff at 15 months.

## 4. Discussion

This is one of the first studies to show the positive impact of a theory-driven report back intervention on taking protective health actions as well as on actual reductions in environmental exposures. This was a longitudinal study of a subsample of homeowners with high levels of radon and/or air nicotine in the home. Each of the 87 hig- risk treatment group participants from the parent RCT [12] received report back based on their actual testing values and their stage of action for radon mitigation and/or adopting a smoke-free home. The current literature on report back provides insight into the clear delivery of written report back materials often sent to participants through the mail [5,9]. This study focused not only on the delivery of verbal information over the telephone, but on the process and how it links to movement along a theoretical health behavior continuum set by the Precaution Adoption Process Model and its stages of action.

The theory-driven report back process was especially effective for those with high radon. Readiness to mitigate for high radon continued to surpass the baseline stage of action over time. In fact, actual radon exposure declined significantly from an average of 7.69 to 3.52 pCi/L in the 15 months following report back. This result was different than the findings of the RCT with the full sample from the parent study, although we controlled for radon and air nicotine risk status in the statistical models [12]. In the RCT, actual radon exposure did not differ significantly either between baseline and 15 months or between the treatment and control groups. This difference in findings may have been due to the fact that this high-risk subsample of participants was more informed, and all of them received report back about the radon values in their homes tailored to their stage of readiness, whereas not all of the treatment participants in the parent RCT elected to receive this intervention (most typically due to lack of baseline testing, precluding the report back intervention). While the general public may be aware of radon, few people test their homes [23] and there are few public health messages about this environmental risk. Public health and health care professionals, including nurses, need to improve access to radon testing [24] in all healthcare settings and use brief problem solving based on readiness for health behavior change to counsel homeowners about high radon levels.

While the theory-driven report back intervention lowered average actual radon values as a group, only 13% of those with high radon mitigated their homes. Among those with high radon who mitigated, all had lowered the radon values in their homes to at or below the EPA action level of 4.0 pCi/L [2,18] by 15 months post report back. Interestingly, the parent grant offered to pay treatment group participants up to 30% of the cost of radon mitigation (up to USD 600) [12], but only a small percentage with high radon values chose to mitigate. Although they had financial assistance, some may not have had enough time or funds to actually mitigate within the study time frame. More research is needed to understand factors contributing to homeowners’ decisions to mitigate for radon.

Tailored report back of high air nicotine levels was effective in increasing stage of action for adopting a smoke-free home within 9 months following report back. However, this effect was not sustained through the 15-month period. Further, the report back intervention did not lower actual exposure to SHS as measured by air nicotine, consistent with the results of the RCT [12]. As in the parent study, we recommend a booster session after 9 months to assist high-risk homeowners to adopt and implement a smoke-free home. Our report back intervention was focused on lowering SHS exposure vs. smoking cessation. Given that tobacco dependence requires intensive treatment [25], future efforts may need to add a smoking cessation segment to the report back intervention including AAR (Ask, Advise, Refer) [26,27], especially if the participant is a current smoker.

One strength of this study is its contribution to the literature; there has been little prior research testing the impacts of theory-driven report back techniques in environmental health science. In addition, this was a relatively large sample of participants who each received tailored information as a result of baseline testing that they performed in their home. One limitation is the lack of comparison group; it was not possible to compare changes over time in stage of action or air quality between those who did and did not receive the report back intervention; nearly all participants who were randomized to the treatment group either completed baseline testing and received report back or they were lost to follow up. Participants in the control group of the parent study did not form a suitable contrast to this evaluation of tailored report back since most of them did not test their homes for radon or air nicotine. The concern of the lack of a comparison group is somewhat mitigated by the longitudinal nature of the design, allowing us to measure the trajectory of stage of action and test values over time among those who received the theory-driven report back intervention after baseline testing. Additional limitations include a regional sample with predominantly White/non-Hispanic, highly educated participants, thus decreasing the generalizability of these findings.

## 5. Conclusions

Access to free test kits coupled with a verbal report back process completed over the telephone can improve readiness to take action to remediate for radon and SHS. A theory-driven report back process reduced radon exposure and promoted the health behavior change necessary to remediate environmental exposure, particularly related to radon mitigation. A report back booster session at 9 months may be necessary to provide more time to mitigate for radon and to provide more intense feedback for those with high levels of SHS exposure. To support lung cancer prevention efforts, public health professionals may consider how home testing for radon and SHS along with a theory-driven report back process can engage patients in lung cancer risk reduction and promote positive health behavior change.

## Figures and Tables

**Figure 1 ijerph-18-10648-f001:**
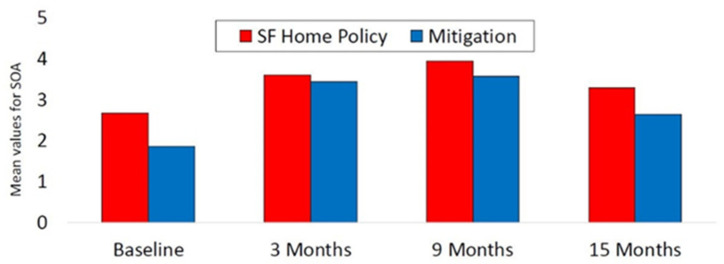
Stage of action among participants with high baseline test values.

**Figure 2 ijerph-18-10648-f002:**
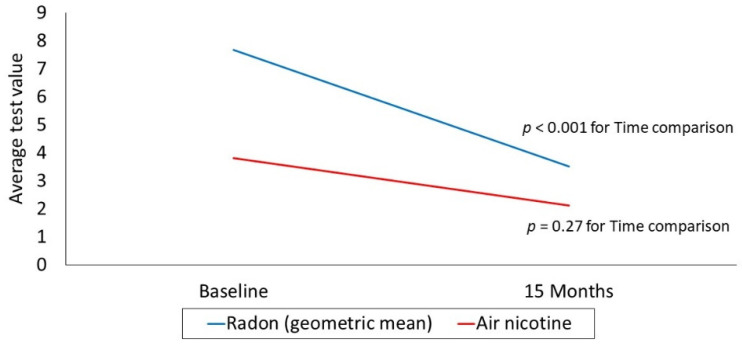
Radon and air nicotine at baseline and 15 months.

**Figure 3 ijerph-18-10648-f003:**
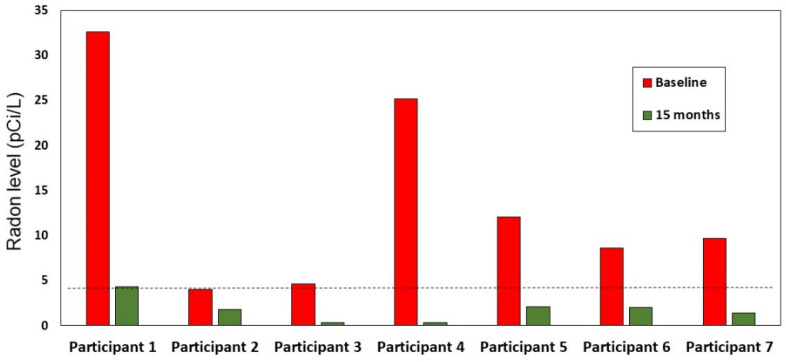
Mitigation radon values at baseline and follow up for the participants who reported mitigating; dotted line denotes EPA action level for radon.

**Table 1 ijerph-18-10648-t001:** Demographic and personal characteristics (*N* = 87).

Characteristic	Mean (SD) or Frequency (Percent)
Age	53.8 years (14.0)
Gender	
Female	55 (63.2%)
Male	32 (36.8%)
Race/ethnicity	
White/non-Hispanic	75 (86.2%)
Other race/ethnicity	12 (13.8%)
College graduate	
Yes	55 (64.0%)
No	31 (36.0%)
Income ≥ USD 60,000	
Yes	47 (54.0%)
No	40 (46.0%)
Smoker(s) in the home	
Yes	44 (50.6%)
No	43 (49.4%)
Family history of lung cancer	
Yes	25 (28.7%)
No	62 (71.3%)
High radon at baseline	
Yes	54 (63.5%)
No	31 (36.5%)
High air nicotine at baseline	
Yes	41 (47.1%)
No	46 (52.9%)

**Table 2 ijerph-18-10648-t002:** Estimates from mixed models for stage of action outcomes and testing results.

	Stage of Action				Testing Values			
	Radon Mitigation (*n* ^1^ = 84)		Smoke-Free Home Policy (*n* = 45)		Radon (*n* = 53)		Air Nicotine (*n* = 64)	
	est. (SE)	*p*	est. (SE)	*p*	est. (SE)	*p*	est. (SE)	*p*
Age	<0.01 (<0.01)	0.10	−0.01 (0.01)	0.25	<0.01 (0.01)	0.15	<0.01 (0.03)	0.78
Male	0.13 (0.15)	0.37	−0.44 (0.31)	0.17	−0.17 (0.18)	0.35	−0.99 (1.11)	0.38
White/non-Hispanic	0.25 (0.19)	0.20	0.76 (0.42)	0.074	−0.19 (0.25)	0.45	−0.20 (1.38)	0.89
College graduate	0.16 (0.16)	0.32	0.31 (0.29)	v.28	0.02 (0.21)	0.93	−2.29 (1.03)	0.034
Family history of lung cancer	0.02 (0.15)	0.87	−0.33 (0.25)	0.18	−0.08 (0.19)	0.67	−0.78 (1.08)	0.47
High value at baseline	−0.94 (0.15)	<0.001	−1.28 (0.31)	<0.001	--	--	--	--
Time Baseline Month 3 Month 9 Month 15	−0.80 (0.16) 0.74 (0.17) 0.88 (0.18) ref	<0.001 <0.001 <0.001 --	−0.37 (0.20) 0.44 (0.21) 0.79 (0.21) ref	0.066 0.039 <0.001 --	0.82 (0.17) -- -- ref	<0.001	1.49 (1.31) -- -- ref	0.27

^1^*n* varies per model due to sporadically missing data.

## Data Availability

No new data were created or analyzed in this study. Data sharing is not applicable to this article.

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
