# Peer review of "Use of Theory-Driven Report Back to Promote Lung Cancer Risk Reduction"

_ijerph, 2021, doi:10.3390/ijerph182010648_

Round 1

Reviewer 1 Report

I reviewed the manuscript "Use of Theory-driven Report Back to Promote Lung Cancer Risk Reduction" by Huntington-Moskos et al.

The study is very interesting and it is well presented, therefore I suggest a minor revision.

I only suggest the possible source(s) of a radon contamination (e.g., environmental? antrophic?).

In addition, deleate the sentence of the MDPI template, which remained in lines 197-198.

Author Response

Thank you for your thoughtful review of the manuscript. Please see attached document for the required point by point response and revisions to the comments provided.

Reviewer 2 Report

The manuscript provides some interesting results from a particular study on lung cancer risk reduction using a report back method on action to improve radon and smoking exposure in the resident houses. There are a few corrections and additions to be made to make the manuscript clearer for the readers. The abstract is difficult to understand, and the "repeated measures mixed models" in the analysis method section are not described in details with different types of covariates. At least references should be given and a simple mathematical equation describing the dependent variable with variates and co-variates is given to allow the reader to understand the change in report back method over time as compared to the baseline.

 Specific comments:

(1) In the abstract: 

Line 12-13: This sentence is incomplete: "Observational one group pre-post design with data collection at 3, 9, and 15 months post-intervention."

Line 14: "... randomized to group and had elevated radon .." should be "...randomized to group that had elevated home radon ..."

Line 17-18: The sentence "Intervention provided free, in-person radon and SHS test kits and a brief telephonic problem solving consultation."  should be clearer "Intervention is provided free such as in-person radon and SHS test kits and a brief telephonic problem solving consultation."

(2) Line 94: "university": please specify which university

(3) Line 112-113: "The brief problem solving was tailored based on the level of exposure based on one of four conditions.." should be "The brief problem solving was tailored based on the level of exposure which is on one of four conditions..."

(4) Line 144: "These repeated measures models//". please provide references for these models. And describe the models as an equation with variates and covariates.

Author Response

(The authors gave the same response as above.)

Reviewer 3 Report

In the study, Free test kits for radon and SHS were distributed to the 87 high risk treatment group participants from the parent RCT, and oral reports were made over the phone about their specific actions to reduce radon and/or adopt smoke-free homes. The effect of the theory-driven report back process on participants with high levels of radon and/or nicotine in the air at home was assessed.

The general idea of this work is very interesting. However, there is no detailed description of the data on test kits for radon and SHS collected and investigated at multiple time points (3, 9 and 15 months), and the corresponding results are poorly described. Thus, in this condition, it is not possible to evaluate the reliability and consistency of results, compromising significantly the quality of the manuscript. Furthermore, the authors did not explain the relationship between the kits test results and the results of each participant's feedback obtained through the theory-driven report back. This represents a critical issue since the authors reported that the theory-driven report back intervention lowered average actual radon values as a group without providing specific feedback content on the impact of the test results. In addition, the study lacked an investigation of the background value of radon in the environment where the participants lived, or whether test Kits for radon and SHS were interfered with by other factors during their use. The lack of explanations of the impact of background values undermines the final results and the conclusions reported in this work. 

 Therefore, due to several relevant weaknesses, I do not recommend this paper for publication.

Author Response

(The authors gave the same response as above.)

Round 2

Reviewer 3 Report

The authors proposed a revised version of the previously rejected manuscript. In the revised manuscript all the computational details have been provided according to first review comments and also the major comment was addressed. However, some major comments have not been addressed satisfactorily. For example, the author did not show the data obtained from the kit test. Moreover, the authors still do not explain the relationship between the kits test results and the results of each participant's feedback obtained through the theory-driven report back. Therefore, considering that many critical issues still characterize the present manuscript, I would be happy to review a revised version of the manuscript after a major revision which addresses.

Author Response

Thank you for your comments. Please see the attached response document for a point by point response. 

Round 3

Reviewer 3 Report

All the revisions have been well made. It can be accepted I think.